# Proteomics Evidence of a Systemic Response to Desiccation in the Resurrection Plant *Haberlea rhodopensis*

**DOI:** 10.3390/ijms23158520

**Published:** 2022-07-31

**Authors:** Petko Mladenov, Diana Zasheva, Sébastien Planchon, Céline C. Leclercq, Denis Falconet, Lucas Moyet, Sabine Brugière, Daniela Moyankova, Magdalena Tchorbadjieva, Myriam Ferro, Norbert Rolland, Jenny Renaut, Dimitar Djilianov, Xin Deng

**Affiliations:** 1Agrobioinstitute, Agricultural Academy Bulgaria, 1164 Sofia, Bulgaria; dmoyankova@abi.bg (D.M.); d_djilianov@abi.bg (D.D.); 2Key Laboratory of Plant Resources, Institute of Botany, Chinese Academy of Sciences, Beijing 100093, China; 3Laboratoire de Physiologie Cellulaire et Végétale, University Grenoble Alpes, CNRS, INRAE, CEA, 38054 Grenoble, France; denis.falconet@cea.fr (D.F.); lucas.moyet@cea.fr (L.M.); norbert.rolland@cea.fr (N.R.); 4Institute of Biology and Immunology of Reproduction, Bulgarian Academy of Sciences, 1113 Sofia, Bulgaria; zasheva.diana@yahoo.com; 5Environmental Research and Innovation (ERIN) Department, Luxembourg Institute of Science and Technology, L-4362 Esch-sur-Alzette, Luxembourg; sebastien.planchon@list.lu (S.P.); celine.leclercq@list.lu (C.C.L.); jenny.renaut@list.lu (J.R.); 6Laboratoire de Biologie à Grande Echelle, Institut de Recherches en Technologies et Sciences pour le Vivant, CEA, Université Grenoble Alpes INSERM, 38054 Grenoble, France; sabine.brugiere@cea.fr (S.B.); myriam.ferro@cea.fr (M.F.); 7Department of Biochemistry, Faculty of Biology, Sofia University, 1164 Sofia, Bulgaria; magdalena@tchorbadjiev.com

**Keywords:** resurrection plant, proteomics, systems biology, subcellular fractionation, drought stress, dehydrin

## Abstract

Global warming and drought stress are expected to have a negative impact on agricultural productivity. Desiccation-tolerant species, which are able to tolerate the almost complete desiccation of their vegetative tissues, are appropriate models to study extreme drought tolerance and identify novel approaches to improve the resistance of crops to drought stress. In the present study, to better understand what makes resurrection plants extremely tolerant to drought, we performed transmission electron microscopy and integrative large-scale proteomics, including organellar and phosphorylation proteomics, and combined these investigations with previously published transcriptomic and metabolomics data from the resurrection plant *Haberlea rhodopensis*. The results revealed new evidence about organelle and cell preservation, posttranscriptional and posttranslational regulation, photosynthesis, primary metabolism, autophagy, and cell death in response to desiccation in *H. rhodopensis.* Different protective intrinsically disordered proteins, such as late embryogenesis abundant (LEA) proteins, thaumatin-like proteins (TLPs), and heat shock proteins (HSPs), were detected. We also found a constitutively abundant dehydrin in *H. rhodopensis* whose phosphorylation levels increased under stress in the chloroplast fraction. This integrative multi-omics analysis revealed a systemic response to desiccation in *H. rhodopensis* and certain targets for further genomic and evolutionary studies on DT mechanisms and genetic engineering towards the improvement of drought tolerance in crops.

## 1. Introduction

At present, climate change, including the expansion of semi-arid regions worldwide and increased drought rates [1,2], has prompted considerable research efforts to understand plant drought responses for applications in crop biotechnology. Recent technological developments and improvements in computer science and the availability of molecular biological data from desiccation tolerant plants have allowed for the design of quantitative experiments and the modeling of systemic responses to drought. 

Water loss leads to changes in cell volume, cell membrane integrity, water potential gradients, solute concentrations, light reactions, and CO_2_ assimilation, which can accelerate the production of reactive oxygen species (ROS). These negative impacts induce carbon starvation and damage various physiological and molecular components, affecting plant survival and productivity. The desiccation tolerance of plants is defined as the ability of their vegetative tissues to withstand cell dehydration (protoplasmic water content <10–20%) [3]. Approximately 300 angiosperm species have been termed “resurrection plants”. These plants have the rare trait to tolerate the almost complete desiccation of their vegetative tissues. Thus, they are appropriate models for studying extreme drought tolerance and elucidating methods to improve crop stress resistance. 

Extensively studied resurrection species have only been found in a few non-closely related angiosperm families, in which the resurrection ability evolved at different times. The Balkan endemic *Haberlea rhodopensis* is a dicot resurrection plant that belongs to the Gesneriaceae botanical family. Similar to drought-sensitive species, decreases in respiration and carbon fixation have been observed in resurrection plants [4,5,6,7,8,9]. Furthermore, ultrastructural differences, mainly related to chloroplasts, have been observed in *H. rhodopensis* and other resurrection plants; such differences include thylakoid rearrangements, the disassembly of photosystem II (PSII) supercomplexes, and a decrease in the cytochrome *b6f* complex during desiccation [10,11,12,13]. However, in *H. rhodopensis*, cyclic electron flow remains active until the plant is almost entirely dehydrated [11], supported by the upregulation of the subunits of the chloroplastic NAD(P)H dehydrogenase complex [14]. A switch between linear electron flow and cyclic electron flow has also been reported in other resurrection plants [15,16], suggesting its function in the drought stress response related to photoprotection and energy supply in resurrection plants. Plants accumulate various enzymes and metabolites involved in the antioxidant defense system to cope with oxidative stress. In *H. rhodopensis*, several superoxide dismutases (SODs) are constitutively transcribed in all organs at increased levels under drought conditions [17]. In addition, different non-enzymatic scavenging systems, such as ascorbate and glutathione, have been reported during drought- and freezing-induced desiccation [18]. Like other resurrection plants, *H. rhodopensis* responds to water loss by accumulating sucrose and raffinose [11,19,20]. The accumulation of sucrose during the later stages of desiccation is likely caused by gluconeogenesis, as indicated by early starch consumption [20,21] and a decrease in glycolytic intermediates as the sucrose content increases [11]. Several classes of drought-protective proteins have been reported in *H. rhodopensis* and other resurrection plants, including late embryogenesis abundant (LEA) proteins and heat shock proteins (HSPs) [9,14,19,22]. Recently, a comparative genomic study of closely related desiccation-tolerant and desiccation-sensitive species in the Scrophulariaceae family revealed that seed-specific and abscisic acid-associated *cis*-regulatory elements in the genome are involved in the expression of LEA proteins in the desiccation-tolerant species *Lindernia brevidens* [23]. Therefore, the expression patterns of LEA proteins in the vegetative tissues of resurrection plants and their specific molecular and cellular mechanisms indicate the similarity of stress responses between drought-tolerant plants and orthodox seeds [24].

Thus far, drought tolerance in *H. rhodopensis* and other resurrection plants has been studied mainly using physiological, transcriptomic, and metabolomic analyses [25,26]. In the present study, we performed large-scale organellar and phosphorylation proteomics of *H. rhodopensis* during desiccation and combined this data with cellular ultrastructure and previously published transcriptomic and metabolomic data [11,14,19,20] with the aim of better understanding what makes this plant extremely tolerant to drought. 

## 2. Results 

### 2.1. Cellular and Subcellular Ultrastructure of Fully Hydrated and Dry H. rhodopensis

*H. rhodopensis* plants can survive desiccation and recover fully upon rehydration. In agreement with this viability, the vast majority of organelles in the fully dry leaves of *H. rhodopensis* were found by transmission electron microscopy (TEM) to be preserved (Figure 1). However, when compared to fresh leaves (Figure 1A), some subcellular structures were rearranged and degraded after dehydration (Figure 1B). Changes in organization were observed, as well as vacuole fragmentation and the relocation of chloroplasts from the periphery to the center of the cell (Figure 1B, left panel). Ultrastructural changes were also observed in chloroplasts and mitochondria. The cristae of some mitochondria were disorganized and reduced (Figure 1B, middle panel), and grana destacking occurred in some chloroplasts (Figure 1B, middle and right panel). Although most organelles were well preserved in the fully dry state, features resembling chlorophagy (phagocytosis in the chloroplast) after degradation were detected (Figure 1B, right panel). Several bubble-like loculi of varying densities and an increased accumulation of plastoglobules were also observed (Figure 1B, right panel).

### 2.2. Quantitative Proteomic Analysis of Fresh and Dry H. rhodopensis Crude Cell Samples

A label-free shotgun quantitative proteomics analysis of fresh and dry *H. rhodopensis* samples led to the identification of 944 significant ID proteins annotated using our in-house RNA-Seq database [14,19] and 638 significant ID proteins using the UniProt Viridiplantae database (Appendix A). Protein hits from the RNA-Seq database were enriched with proteins with significant changes identified in the database according to the corresponding UniProt identifiers and analyzed further. The distribution of replicates from the fresh and dry states in the first three components of PCA according to the abundances of identified proteins is shown in Figure 2A. According to Gene Ontology (GO) annotation, the identified proteins were mainly associated with chloroplasts, mitochondria, non-membrane-bound organelles, plastids, vacuoles, and the nucleolus (Figure 2B). Clusters containing proteins with similar mean log_2_ fold-change values are visualized in a heat map in Figure 2C. 

We attempted to annotate the significantly upregulated or downregulated proteins (|log_2_ fold change| > 1); however, 20 protein sequences with significant changes could not be annotated by matching their de novo assembled RNA contigs from *H. rhodopensis* to current databases. The greatest desiccation-induced downregulation was found for the following proteins: proteasome subunit alpha type-6, ATP synthase subunit beta, gibberellin receptor GID1, chloroplast ATP-dependent zinc metalloprotease, plastid lipid-associated protein, metacaspase-4, auxin-induced in root cultures protein 12, auxin-binding protein ABP19a, chloroplast triosephosphate isomerase, a PSII 10 kDa polypeptide, the mitochondrial ADP/ATP carrier protein, 26S protease regulatory subunit 8 homolog A, and a thylakoid luminal 29 kDa protein. The significantly upregulated proteins (log_2_ fold change > 1) included various LEA proteins, HSPs, thaumatin-like proteins (TLPs), early light-inducible proteins (ELIPs) involved in cellular protection, and antioxidant enzymes (e.g., Cu/Zn SOD, L-ascorbate peroxidase, and monodehydroascorbate reductase). The levels of the V-type proton ATPase subunit, PSI reaction center subunit, PetA protein, vacuolar H^+^-ATPase catalytic subunit, translocator protein homolog, and chromatin remodeling complex subunit were also significantly increased under drought conditions. Several upregulated enzymes involved in sugar metabolism were identified, including sucrose synthase, triosephosphate isomerase, glyceraldehyde-3-phosphate dehydrogenase, UDP-apiose/xylose synthase, and a member of the transketolase protein family.

### 2.3. Subcellular Fractionation and Gel-Based Proteomics of Crude, Organelle, Phosphoprotein, and Low-Abundance Protein Fractions

To avoid the significant difficulties related to the purification of intact chloroplasts from dried leaves, as suggested by our TEM observations (Figure 1) and previous reports [27], we used a 2D-DIGE gel-based approach. By using 2D-DIGE technology for accurate intra and intergel matching and comparisons, we matched several 2DE proteome catalogs from crude cell extracts, purified organelles, and phosphorylated and low-abundance fractions enriched via affinity purification and combinatorial hexapeptide ligand library treatment from both fresh and dry plants with an internal standard and pI/MW (BioRad) marker (Appendix A). This allowed us to analyze each spot in each population of gels normalized to an internal standard and pI/MW and to match organelle proteomes from fresh leaves with desiccation-induced changes in the crude leaf proteome, phosphoproteome, and low-abundance proteome.

We were able to purify intact chloroplasts and increase the yield of mitochondria from the fresh leaves of *H. rhodopensis* after optimizing previously published purification protocols [28]. Because *Haberlea* chloroplasts are denser than spinach chloroplasts, they tend to accumulate at the bottom (i.e., with the nuclei and cell debris) of the classic Percoll gradients used to purify spinach [29] and *A. thaliana* [30] chloroplasts (Appendix A). To limit cross-contamination, we established a modification using a 90% Percoll cushion to condense the intact chloroplasts in the layer near the bottom of the tube (Figure 3A, left panel), rather than a 40–80% Percoll gradient. The chloroplasts were then fractionated (Figure 3A, middle panel) into stroma, envelope, and thylakoids, according to the procedure described by Salvi et al. (2008). Next, to optimize our previously published protocol [28], we used rate-zonal centrifugation and step gradients to purify the mitochondria. This procedure allowed for the separation of two bands containing mitochondria, as described previously [31]. The denser and more abundant fraction (Figure 3A, right panel) was used for further analyses. To evaluate the level of cross-contamination in organelle subcompartments after purification, proteins from the chloroplast and mitochondrial subcompartments were resolved using sodium dodecyl sulfate-polyacrylamide gel electrophoresis (SDS-PAGE) (Figure 3B, upper panel). Then, they were analyzed by immunoblotting to assess the presence of specific organelle markers (Figure 3B, lower panel). Compared with the crude cell extract, the chloroplast and stromal fractions were slightly enriched in RbcL—a stromal marker—whereas weak or no signals were detected in the thylakoid, envelope, and mitochondrial fractions. Similar results were obtained using an anti-LHCP (thylakoid marker) antibody, which detected target proteins only in the crude extract, chloroplast fraction, and thylakoid fraction, rather than in the stromal, envelope, or mitochondrial fractions. The chloroplast envelope marker E21 was only detected in the purified envelope fraction (not in the crude cell extract), indicating a strong enrichment (>100) that resulted from the purification of this fraction. Finally, TOM40—a mitochondrial marker—was detected in the mitochondrial fraction, where the signal was enriched relative to the signal from the crude cell extract. 

Further, by applying the organelle proteomes together with the crude leaf protein extracts, phosphoproteins, and enriched low-abundance proteins, we were able to create a comprehensive 2D proteome map of *H. rhodopensis* that showed 1253 spots matched to the total leaf proteome, showing a 40% increase in the number of detected spots when compared with the total leaf extract (Appendix A and Figure 3C). Moreover, this 2D gel-based approach allowed for the de novo sequencing of peptides derived from spots. Spots that presented significant abundance or phosphorylation changes in the dry state and some selected spots from the organelle fractions were analyzed using matrix-assisted laser desorption/ionization time-of-flight/time-of-flight mass spectrometry (MS). Of the 73 spots analyzed, 26 were identified (Appendix A), and 16 resulted in single-hit identifications (Table 1). Desiccation induced the overexpression and phosphorylation of a stem-specific protein (spot 3), as well as the overexpression of polyphenol oxidase (PPO), localized in the stroma (spot 26). Several proteins linked to carbon fixation and photosynthesis were detected in different fractions. RuBisCO activase and RbcL (spots 11 and 18, respectively) decreased in desiccated plants. Chlorophyll *a-b* binding protein 40 (spot 2) was overexpressed, exhibiting increased phosphorylation in the phosphoproteome. The level of oxygen-evolving enhancer protein 1 (OEE1) increased in the dried leaves’ total proteome and phosphoproteome (spot 12). Among the stress-responsive proteins, a drought-induced increase in phosphorylated dehydrin (pI 5.9, MW 20 kDa; spot 20) was detected in the thylakoid fraction. Several proteins could not be annotated using the databases employed in this study, indicating a potential for further research.

### 2.4. Expression, Posttranslational Modifications, and Subcellular Localization of Dehydrins

The protective role of dehydrins in abiotic stress tolerance prompted us to validate the 2D-DIGE results and examine *H. rhodopensis* dehydrins in more detail. Based on the multiplexed visualization of 2D immunoblots of the internal control, we found two isoforms of dehydrin with estimated pIs of 5.9 and 6.31 and a MW of 20 kDa (Figure 4A, green spots); the more acidic isoform corresponded to the phosphorylated dehydrin identified in the DIGE analysis (Figure 4C, spot 20; Table 1). 

The spots, marked as Dhn1 and Dhn2, were matched to the corresponding members of the crude proteome according to the DIGE internal standard (Figure 4B). Each detected protein–spot couple was excised and analyzed using LC-MS/MS (Appendix A). The analyzed spots matched the dehydrin-like protein Dh2 (*Paraboea crassifolia*; accession number AAO86690.1). The translation of the RNA contig and amino acid sequence alignment with the sequence of Dh2 from *Paraboea crassifolia* showed that the dehydrin from *Haberlea* belonged to the YSK_2_ type dehydrins (Appendix A). Consistent with previous results (Figure 2C, Table 1), the phosphoprotein Dhn1 accumulated at higher levels in response to desiccation. A quantitative analysis showed a more than two-fold increase in the abundance of Dhn1 compared to its level in fully hydrated plants (Figure 4B, lower right panel). The change in the abundance of the non-phosphorylated Dhn2 isoform was inversely proportional to that of Dhn1; drought stress induced a 1.6-fold decrease in Dhn2. Furthermore, the ratio of phosphorylated isoforms to non-phosphorylated isoforms was lower in fresh plants than in dry plants. The dynamics of dehydrin expression at different states of desiccation showed one major band with an estimated molecular mass of 20 kDa in SDS-PAGE using an anti-dehydrin antibody, in agreement with the DIGE data. No significant changes in protein abundance were observed throughout the desiccation process (Figure 4C, arrow). The immunoblots of the chloroplast fraction from fresh plants and the thylakoids of both fresh and dry leaves confirmed the localization of dehydrin in thylakoids, consistent with the 2D-DIGE findings. The immunoblots also showed an increased signal in the thylakoids of dry plants compared to fresh plants (Figure 4D). 

### 2.5. Pathway Analysis and Co-Expression Changes of the Proteome, Transcriptome, and Metabolomics of H. rhodopensis

We combined RNA-Seq data from previously published transcriptomic studies [14,19] of fresh and desiccated *H. rhodopensis* plants and used RNA contigs to identify proteins for integrative network analysis. All annotated genes identified using the RNA-Seq and proteomics data were enriched and functionally grouped in a network via ClueGO analysis in processes related to cell wall organization, organelle organization, photosynthesis, peroxisome organization, primary and secondary metabolism, autophagy, programmed cell death, oxidative stress, DNA metabolic processes, stress response, and seed development (Appendix A). To directly compare proteome and transcriptome data, the averaged log2 transformed fold changes between dry and fresh plants of all identified proteins were plotted with the expression changes of their corresponding contigs (Figure 5A). Significant expression changes with a log_2_FC (fold change) > 0.5 in proteins and their corresponding contigs are represented by different colors. About 40% of the expression levels of proteins corresponded to changes in their mRNA levels (data not shown), while other genes showed the up/downregulation of transcription with no associated significant changes in the accumulation of proteins. Genes involved in the responses to desiccation, heat, and light intensity; photoprotection; glyoxylate and the tricarboxylic acid cycle; oligosaccharide metabolic processes; and seedling development exhibited mRNA expression levels that were consistent with an increased accumulation of their corresponding proteins under desiccation. Conversely, genes involved in PSII assembly, cytoskeleton organization, nucleoside salvage, and porphyrin-containing metabolic processes exhibited decreased mRNA expression levels in tandem with lower levels of the corresponding proteins. Genes that showed increased transcription but decreased protein levels were involved in calcium ion binding, translation elongation, and the defense response. In contrast, several genes induced by biotic stress, protein import into the nucleus, and photosynthesis regulation showed a slight decrease in their transcripts and an increase in protein levels (Appendix A).

We also used the available metabolomics data [11,19,20] to identify significant metabolic pathways by employing the KEGG database to calculate pathway activity perturbations (Appendix A and Figure 5B,C). In the figures, proteomics changes are represented by their gene names. The accumulation of transcripts and metabolites was found in several different processes and pathways, including the light reaction of photosynthesis and C-4 fixation, detection of stimulus, cell wall and organelle organization, antioxidant activity, DNA metabolite processes, response to stress, seed development, flavonoid biosynthesis, lipid degradation, ER stress, autophagy, and cell death. Several queries exhibiting the up/downregulated co-expression of mRNA and protein levels were not annotated (NA) in the current databases and represent candidates of interest for further research. We used the InterProScan tool [32] to examine the domains in these queries. In dry plants, downregulation was observed for proteins with domains like the transmembrane helix and cytoplasmic domain (NA1), iron_permease_Fet4-like domain (NA2), and ATP-dependent RNA unwinding (NA3). Upregulation was observed in proteins with domains similar to the disordered tumor suppressor-like domain (NA4), glycoside hydrolase family domain (NA5), and lipoprotein A-like domain (NA6) (Figure 5B). According to the pathway flow calculations on primary metabolism, signal propagation increased at several nodes in the network topology (Figure 5C). Decreases in the fatty acid and branched amino acid pools, as well as increases in peroxisomal acyl-coenzyme A oxidase 1 (ACOX1) transcription and dihydrolipoamide branched chain transacylase E2 (DBT) protein expression, were significantly associated with the signal accumulation representing succinate dehydrogenase complex flavoprotein subunit A (SDHA) gene overexpression. The transfer of reducing equivalents from mitochondria to the cytosol for the de novo synthesis of sugars was supported by a decrease in malate and the overexpression of pyruvate kinase (PK), indicated by an increase in signal perturbation. Additionally, reductions in CO_2_ fixation and RbcL protein level implied a decrease in flow through the Calvin cycle. In contrast, the accumulation of glyceraldehyde-3-phosphate dehydrogenase (GAPDH) and phosphofructokinase A (PfkA) proteins (indicated by increased signal propagation at these nodes) represented the flow of triose-phosphates derived from mitochondria. At the end of the constructed pathway, the increase in sucrose synthase (SUS) protein level (indicated by increased signal flow) affected sucrose accumulation at the sink node.

## 3. Discussion

In the present study, we explored changes in the proteome of the resurrection plant *H. rhodopensis* upon desiccation for the first time using shotgun and gel-based approaches coupled with transmission electron microscopy observations of subcellular organization. For the in-depth proteome analysis, we evaluated the changes in protein accumulation, in addition to their organelle localization and phosphorylation. We applied various extraction and enrichment protocols to obtain proteins and organelles from recalcitrant, phenol-rich plant tissues. The integration of these organelle- and phosphoproteomics-based investigations with transcriptomics and metabolomics data enables the construction of more knowledge-rich and accurate networks of stress responses (Figure 5, Appendix A). 

Similar to previous findings in *H. rhodopensis* and other resurrection plants [33,34,35], the transmission electron microscopy micrographs did not reveal significant cellular damage in most cells from the desiccated leaves of *H. rhodopensis* (Figure 1). On the other hand, we detected changes in the cell wall proteins (Figure 5B). Cinnamoyl-CoA reductase (CCR1), involved in lignin synthesis during various developmental stages [36], was downregulated at the mRNA and protein levels under stress. The transcription of the gene encoding cellulose synthase also decreased (CESA6). However, the protein levels of UDP-apiose/xylose synthase (AXS1) and pectin esterase (PME2.2) increased upon desiccation (Figure 5B). The increased PME2.2 protein level in dry *H. rhodopensis* is consistent with previous results obtained for the resurrection plants *Craterostigma wilmsii*, *C. plantagineum*, and *L. brevidens* showing de-methylesterified pectin accumulation in the cell wall during desiccation [37]. Cell wall architecture and plasticity depend on cross-linking between a rigid cellulose-xyloglucan network and pectin polysaccharides. This linking is facilitated by rhamnogalacturonan II (RG-II) [38]. Changes in RG-II and pectin composition lead to more rigid cell walls upon dehydration in resurrection plants [39,40]. 

Desiccation changes cellular and organelle organization, inducing vacuole fragmentation and chloroplast relocation from the cell periphery to center in *H. rhodopensis* (Figure 1) and other resurrection plants [33,35]. We observed a pronounced decompactization of the grana stacks in dry chloroplasts (Figure 1), although well-preserved and compacted grana stacks were predominant, ensuring the rapid recovery of photosynthesis after rehydration. Thylakoid remodeling in *H. rhodopensis*, *Boea hygrometrica*, and *Craterostigma plantagineum* reportedly causes changes in the photosynthetic machinery and performance [10,11,12,13]. Various phosphorylation events are involved in thylakoid membrane structure regulation, antenna remodeling, and state transitions in *A. thaliana*, including the light harvesting complex subunit LHCP (CAB40), which is involved in the capture and delivery of excitation energy to the photosystems in *A. thaliana* under water stress [41,42,43]. Here, we found an increased phosphorylation of LHCP and OEE1 (Table 1, Figure 5B) in dry *H. rhodopensis*, suggesting a common role of this posttranslational modification in PSII supercomplex organization in both desiccation-tolerant and desiccation-sensitive species. PSII functionality also depends on the state of the oxygen-evolving complex, which is related to the expression of manganese-stabilizing proteins [44]. We found that OEE1 was also overexpressed in dry plants (Table 1, Figure 5B). This protein is involved in PSII assembly and activity in higher plants; it has been reported that OEE1 activity is regulated by phosphorylation via the AtGRP-3/WAK1 signaling pathway [45,46]. The GRP/WSK1 signaling pathway was also identified in the resurrection species *C. plantagineum* [47]. It would be interesting to know if OEE1 is also regulated by phosphorylation via the AtGRP-3/WAK1 signaling pathway in *H. rhodopensis*. The PSI reaction center subunit L (PsaL) was overexpressed in dry *H. rhodopensis* (Figure 5B), as was previously observed in *C. plantagineum* [9], indicating the importance of maintaining PSI functionality in resurrection plants during desiccation. Moreover, during drought stress in *H. rhodopensis*, we found the accumulation of the petA gene product—also identified in the genome of *Boea hygrometrica* [48]—a component of the cytochrome b6-f complex, which mediates cyclic electron flow around PSI and state transitions [49]. We also detected PPO in the stromal fraction, matching spots that indicated overexpression in the total proteome during desiccation (Figure 3B and Figure 5); this protein also matched two other isoforms with multiple hits (Appendix A). In good agreement with these observations, the abundance and activity of PPO also increased in the leaves of *B. hygrometrica* and *Craterostigma plantagineum* upon desiccation [9,50]. PPOs catalyze the oxygen-dependent oxidation of phenols to quinines. However, their functions in chloroplasts, particularly their roles in the oxidative stress response and plant defense against environmental stress, have not been fully elucidated [51]. 

In the dry leaves, we also detected cristae disorganization in some mitochondria (Figure 1), which implied an overall decrease in energy metabolism [11]. The protein level of the mitochondrial outer membrane protein porin (VDAC) was increased in dry *H. rhodopensis* (Figure 5B). VDACs have been identified in fungi, plants, and metazoans, including invertebrates. Presumably, similar evolutionary processes occurred across these lineages. These porins are involved in the exchange of numerous ions and molecules and the regulation of stress responses [52]. The decrease in energy metabolism is likely correlated to slowing down the physiological activities in dry leaves and energy saving to support the period of desiccation tolerance. 

The transcription of several factors linked to peroxisome biogenesis (enriched according to the RNA-Seq library) was increased in dry *Haberlea*, implying the enhanced assembly and maintenance of peroxisomes (Figure 5B). Plant peroxisomes are involved in antioxidant activity and embryogenesis and seedling growth, and these processes are fueled by fatty acid β-oxidation [53]. By incorporating available omics data, we were able to develop a model that linked decreases in fatty acid and branched amino acid pools with the accumulation of sucrose (Figure 5C), the most abundant osmoprotectant in *H. rhodopensis*, as a response to the inhibition of carbon fixation [5,11,20]. According to the pathway flow calculations, signal propagation increased at several nodes, suggesting that flow control presumably depended on the overexpression of ACOX1, DBT, and SDHA (Figure 5C), which led to the production of substrates for de novo sucrose synthesis involving glycolytic intermediates.

Our electron micrographs showed occasional damage in the cellular structures, the degradation of chloroplasts, and the development of chlorophagy-like features (Figure 1), which were previously described in *Craterostigma pumilum* [54]. Furthermore, transcripts associated with autophagy (Atg3, Atg8C, and Atg9) and proteins involved in the ER-stress response (Erd2) increased during drought stress in *H. rhodopensis* (Figure 5B). The overexpression of genes related to autophagy and ER stress, which engage the unfolded protein response and subsequently activate autophagy, has been reported in desiccation-tolerant *Boea hygrometrica* [55,56]. Autophagy promotes or inhibits cell death depending on the internal and external environment and cell type. It is mainly induced during development, environmental stress, and nutrient deficiency; it recycles and generates substrates for energy metabolism and represents a process of interest for improving crops [57,58]. The turnover and processing of damaged structures (e.g., lipids, proteins, and organelles) enables the production of raw materials for the de novo synthesis of various molecules, even in the absence of carbon fixation under drought stress. Despite the observations of partial cellular damage and the development of autophagy-like features, we observed the full recovery of the desiccated plants upon rehydration, suggesting an anti-cell death function of autophagy activation. Programmed cell death occurs in multicellular organisms and is subjected to genetic control; it depends on two types of metacaspases found in protozoans, plants, fungi, and prokaryotes [59]. The activation of autophagy mechanisms in the dry leaves of *H. rhodopensis* is in agreement with the decreasing transcription and protein accumulation of AMC4, a positive regulator in biotic and abiotic stress-induced programmed cell death in *H. rhodopensis* (Figure 5B), which suggests that cell death is avoided in *H. rhodopensis* when subjected to severe drought stress in contrast to non-resurrection plants [60]. Similarly, *Tripogon loliiformis*, a desiccation-tolerant grass from Australia, induces autophagy to suppress senescence and cell death during desiccation; a lesser accumulation of transcripts is associated with apoptosis and senescence, while more abundant autophagy-related transcripts are observed throughout dehydration and desiccation [61]. 

In *H. rhodopensis*, six SOD isoforms have been identified and classified from seven SOD gene family clones via transcriptome analysis and cloning [17]. In agreement with their upregulated transcription and activity [18,19], we detected the accumulation of the Cu/Zn SOD proteins during drought stress. The higher accumulation of glutathione peroxidases and ascorbate peroxidase observed in desiccated *H. rhodopensis* confirms the well-known functions of these enzymatic reactive oxygen species scavengers in plants, including *H. rhodopensis* [18,62]. Lactoylglutathione lyase (DJ-1 homolog) was overexpressed in dry *H. rhodopensis* (Figure 5B). In general, the DJ-1 protein is involved in the ER stress response; it is classified as an antioxidant protein that exhibits a considerable similarity to human DJ-1 and its *A. thaliana* homolog [63]. The overexpression of this protein in *Arabidopsis* increases its tolerance to various environmental conditions by decreasing oxidative stress via interactions with SOD1 and glutathione peroxidase 2 [64]. The accumulation of proteins linked to both stress responses and seed development was observed in this study (Figure 5B). Such proteins included ELIPs, HSPs, TLPs, embryonic protein DC-8, LEA protein D-29, LEA protein D-34, desiccation protectant protein LEA14 homolog, and desiccation-related proteins. The roles of HSPs, LEAs, and ELIPs in the drought stress response in resurrection plants have been comprehensively discussed elsewhere [9,24,25,26,65]. LEA proteins are distributed in distinct subcellular compartments and contribute to the stability of intracellular membrane structures. Our observations are in agreement with the drought-induced and constitutive expression of LEA genes reported in other resurrection plants [14,19,22]. In addition, we also observed the drought-induced and constitutive expression of another class of disordered proteins with chaperone functions—TLPs—in *H. rhodopensis* [66,67].

Dehydrins are highly hydrophilic proteins that belong to the group II LEA proteins. Their protective role in the plant stress response has been well documented [67,68]. In the present study, we found a phosphorylated (Dhn1) and a non-phosphorylated (Dhn2) isoform of a 20 kDa thylakoid-located dehydrin (Figure 4 and Figure 5B). RNA-Seq and MS2 data showed the similarity of the identified protein to Dh-like dehydrin from the resurrection plant *Boea crassifolia* [69], which belongs to the YSK_2_-type dehydrins (Appendix A). The anti-dehydrin antibody recognized a 20 kDa protein present in the control and during all stages of desiccation due to an equal net amount of both isoforms, indicating its constitutive expression. Gechev et al. (2013) found the abundant expression of some of *H. rhodopensis* LEAs and HSPs in well-watered control plants. They suggested that the transcriptome of this species might be primed for desiccation tolerance. Interestingly, we detected a significantly increased phosphorylation of dehydrin upon desiccation, which suggests the involvement of this posttranslational modification in the desiccation tolerance priming of *H. rhodopensis*. The increasing levels of dehydrin phosphorylation and its accumulation in dried thylakoids suggest that phosphorylation might affect the transport of dehydrin into chloroplasts. Recently, a phosphoproteomic study showed that SNF1-related protein kinases phosphorylate the dehydrins ERD10 and ERD14 in response to stress, implying that ERD14 phosphorylation within the S segment helps to regulate dehydrin subcellular localization in *A. thaliana* in response to stress [70]. The overexpression and phosphorylation of an SNF1-related kinase detected in the present study using 2D-DIGE analysis indicates its putative involvement in dehydrin phosphorylation. However, further analysis is needed to comprehensively elucidate the phosphorylation and transport of thylakoid-located dehydrin in response to desiccation in *H. rhodopensis*.

## 4. Materials and Methods

### 4.1. Plant Material and Desiccation Stress

*Haberlea rhodopensis* plants were propagated in vitro and then transferred to soil [71]. The plants were grown in well-watered pots in a controlled environment at 24 °C with a 16:8 day and night photoperiod, 40% to 60% relative air humidity, and a photon flux density of 36 µmol m^2^·s^−1^ for about a year. Dehydration was imposed by withholding water, as described by Mladenov et al. [11].

### 4.2. Transmission Electron Microscopy (TEM)

Fully hydrated and fully dry leaf samples were quickly washed and wiped to remove any soil before they were fixed in 0.1 M PB (pH 7.4) containing 2.5% (*v*/*v*) glutaraldehyde for 2 h at room temperature and stored overnight at 4 °C. The leaves were then washed five times in 0.1 M PB (pH 7.4) before being fixed by a 1 h incubation on ice in 0.1 M PB (pH 7.4) containing 2% osmium and 1.5% ferricyanide potassium. After being washed five times with 0.1 M PB (pH 7.4), the samples were resuspended in 0.1 M PB (pH 7.4) containing 0.1% (*v*/*v*) tannic acid and incubated for 30 min in the dark at room temperature. The leaves were washed five times with 0.1 M PB (pH 7.4), dehydrated in ascending sequences of ethanol, infiltrated with an ethanol/Epon resin mixture, and finally embedded in Epon. Ultrathin sections (50–70 nm) were prepared with a diamond knife on a PowerTome ultramicrotome (RMC products) and collected on 200 µm nickel grids. The ultrathin sections were examined on a Philips CM120 transmission electron microscope operating at 80 kV [71].

### 4.3. Shotgun Proteomics

Total protein extracts from fresh and dry plants (prepared as described above) were analyzed in triplicate using LC-MS/MS. For each sample, 45 µg of total proteins was loaded into SDS-PAGE until separation on the gel was reached. Stacked protein bands were excised, and the proteins were reduced, alkylated, and destained prior to digestion with trypsin enzyme (sequencing mass grade, Promega). The extracted peptides were analyzed with an Ekspert nanoLC™ 425 system (Eksigent, SCIEX, Redwood City, CA, USA) coupled to a TripleTOF^®^ 6600 MS (Sciex) [9]. Peptide identification was carried out by searches in a homemade *Haberlea* RNA-Seq contigs database (1605312 seq) and *Boea* RNA-Seq contigs database (68670 seq) [14,19,55] and the UniProt *Viridiplantae* database released in March 2019 (7072838 sequences), as recently described in [9]. The MS proteomics data were deposited to the ProteomeXchange Consortium via the PRIDE [72] partner repository with the dataset identifier PXD030496.

### 4.4. Purification of Chloroplasts, Mitochondria, and Chloroplast Subcompartments

The simultaneous preparation of chloroplasts and mitochondria was performed according to Mladenov et al. [28] with some modifications. Briefly, 75 g of fresh leaves was harvested and homogenized three times for two seconds each in a Warring blender containing 50 mL of ice-cold grinding buffer (0.33 M sucrose, 10 mM Tricine, 30 mM MOPS-KOH pH 7.9, 5 mM EDTA, 10 mM NaHCO_3_, 2 mM MgCl_2_, 0.1% (*w*/*v*) BSA, 2 mM DTT, 20 mM ascorbate, 1% (*w*/*v*) PVP40). The homogenate was filtered through nylon mesh (20 μm pore size) and three layers of Miracloth and then centrifuged at 1500× *g* for 6 min at 4 °C. The pellet containing crude chloroplasts was gently resuspended (using a small paintbrush) in chloroplast washing buffer [73], loaded onto a three-step Percoll gradient consisting of 90%, 60%, and 40% Percoll layers in washing buffer, and then centrifuged at 4000× *g* for 10 min at 4 °C. The green bands from the 40–60% and 60–90% interfaces were collected and washed twice in washing buffer using two centrifugations at 1500× *g* for 6 min at 4 °C. To disrupt intact chloroplasts, the pellet was resuspended in hypotonic buffer [73] and then stored in liquid nitrogen. The subsequent chloroplast fractionation into stroma and thylakoids was performed using a sucrose gradient according to Salvi et al. [29]. The supernatant from the first centrifugation (1500× *g* for 6 min at 4 °C, see above) was centrifuged again at 6000× *g* for 5 min at 4 °C to pellet the residual part of the broken chloroplasts. Then, the supernatant from this second centrifugation was centrifuged again at 20,000× *g* for 10 min at 4 °C. The mitochondria from the pellet were then isolated using a three-step Percoll gradient (80%-33%-20%) according to Lang et al. [31]. The purity of the obtained fractions was monitored by Western blot analyses using antibodies raised against different protein markers: a translocator of the outer mitochondrial membrane 40 (TOM 40), light-harvesting complex proteins (LHCPs) of the thylakoid membrane, the large subunit of RuBisCO (RbcL) from the chloroplast stroma, and the E21 chloroplast envelope protein 21 (E21). The antibody raised against LHCPs from *Chlamydomonas reinhadtii* [74] was provided by Dr. Olivier Vallon (Institut de Biologie Physico-Chimique, Paris, France) and used at a 1/20,000 dilution. The anti-TOM40 antibody (formerly MOM42) [75] recognizes an outer membrane protein from *Vicia faba* mitochondria (used at a 1/2000 dilution). The two antibodies raised against the chloroplast envelope protein E21 and the large subunit of RuBisCO from *Spinacia oleracea* were provided by Dr. Maryse Block (Laboratoire de Physiologie Cellulaire & Végétale, Grenoble, France) and used at a 1/2000 dilution. The Western blot analyses were performed using Amersham™ ECL™ Western Blotting Detection Reagents (Amersham).

### 4.5. Purification of Phosphoproteins and Low-Abundance Enriched Proteins and Non-Targeted Gel-Based Proteomics

Total leaf proteins from fresh and dry plants were extracted using phenol/sodium dodecyl sulphate extraction, essentially according to Wang et al. [76], with three washing steps for the protein pellets with 0.3 M Guanidine hydrochloride in ethanol and one with absolute ethanol. Crude leaf proteins from fresh and dry plants were enriched for phosphoproteins and low-abundance proteins. Phosphoprotein isolation was performed via affinity chromatography using the Pro-Q^®^ Diamond Kit (Invitrogen, Waltham, MA, USA) according to the manufacturer’s instructions. The enrichment of low-abundance proteins was achieved using a combinatorial peptides ligand library (CPLL), ProteoMiner (BioRad, Hercules, CA, USA), according to the manufacturer’s instructions. Three biological replicates were used for each extraction and purification method.

For non-targeted gel-based analysis, 75 μg of proteins from crude extracts, phosphoproteins, low-abundance enriched proteins, and organelle fractions were cleaned up using a ReadyPrep 2-D kit (BioRad, Hercules, CA, USA) and labeled with fluorescent dyes (GE Healthcare). They were subsequently separated using two-dimensional difference gel electrophoresis (2D-DIGE) according to the manufacturer’s instructions. Biological replicates (*n* = 3) with dye exchange between the samples from fresh and dry plants were used for the statistical analyses. The samples were focused on an Ettan IPGphor3 Isoelectric Focusing Unit with a total amount of 60,000 Vh on Immobiline^®^ DryStrips (IPG strips) 18 cm, pH range 4–7 (GE Healthcare, Chicago, IL, USA). Proteins were then separated onto the second dimension of sodium dodecyl sulphate-polyacrylamide gel electrophoresis (SDS-PAGE) with a Ruby electrophoresis unit (GE Healthcare) using 8–16% gradient gels prepared in a multiple gel caster. Subsequently, the gels were visualized with a Typhoon (GE Healthcare, Chicago, IL, USA) and analyzed using the Image Master 2D Platinum 7.0 software (GE Healthcare, Chicago, IL, USA). The gel images were organized in match sets (Appendix A), and spots from all protein fractions of the fresh and dry states were matched and quantified. The intensity of each spot was first processed by background subtraction, and volume % was used for DIGE quantification. To fine-tune spot detection, a threshold of volume % > 0.006 was set. Changes in protein expression were determined as the fold-change ratio between the fresh and dry plants, and spots that exhibited an increased or decreased fold change above 1.5, a *p* value < 0.05, and/or specific organelle localization were listed and targeted for excision by an Ettan™ Spot Picker (GE Healthcare, Chicago, IL, USA). Proteins from the gel plugs were digested, extracted, and spotted onto an MALDI plate by the Tecan modular robotic pipetting platform EVO2 (Tecan Trading AG, Männedorf, Switzerland) following a homemade protocol. Matrix-assisted laser desorption/ionization time-of-flight/time-of-flight mass spectrometry (MALDI-TOF/TOF) analysis was performed using a TOF/TOF™ 5800 (AB SCIEX, Redwood City, CA, USA) mass spectrometer in MS and MS/MS mode. Database interrogation was carried out with the ProteinPilot software v4.5 (AB Sciex) on an in-house Mascot server version 2.6.1 (Matrix Science Ltd., London, UK). Spectra were searched against the homemade library described above and the SwissProt protein database (2019_11, 553,231 sequences) for *Viridiplantae.*

### 4.6. Targeted Gel-Based Proteomics

For the targeted analysis of dehydrins, we performed Western blots on crude protein extracts and chloroplast and thylakoid fractions after SDS PAGE or 2D IEF/SDS PAGE separation. Considering the difficulties related to the extraction of intact chloroplasts from fully dry leaves, we applied different gradients for the purification of thylakoids from fresh and dry plants. The same buffers, grinding, washing and centrifugation conditions were applied as described above; however, instead of the three-step Percoll gradient, a two-step Percoll gradient consisting of 17% and 90% cushions was used. Changes in the isoelectric point/molecular weight (pI/MW) of dehydrins were evaluated via 2D-DIGE Western blot using crude protein extract mix from fresh and dry plants (internal standard) pre-labeled with Cy3 (GE) and separated on 2D-PAGE simultaneously with the Cy2 (GE) pre-labeled 2D pI/MW marker (BioRad, Hercules, California, U.S). Subsequently, the proteins were transferred to a nitrocellulose membrane and probed for dehydrins with a primary rabbit polyclonal antibody raised against a synthetic peptide representing a highly conserved domain (the K segment) of dehydrins (kindly provided by Prof. Timothy Close) at a 1:1000 dilution and a Cy5-labelled anti-rabbit IgG (dilution 1:2000) secondary antibody (GE Healthcare, Chicago, Illinois, United States). The membranes were scanned with Typhoon, and the pI/MW of the spots corresponding to dehydrin signals were calculated according to the pI/MW marker and matched to the internal standard with the Image Master 2D Platinum software using the DIGE algorithm. Afterward, the corresponding 2D spots from fresh and dry plants were quantified in 2D-DIGE gels and subsequently analyzed by liquid chromatography coupled to a mass spectrometer (LC-MS/MS). Briefly, the 2D gel spots were excised, proteins were extracted from the gel pieces and digested, and peptides were analyzed using online nanoLC-MS/MS (Ultimate 3000, Dionex, Sunnyvale, California, USA and LTQ-Orbitrap Velos Pro, Thermo Fischer Scientific, Waltham, Massachusetts, U.S) as described previously [77]. Peak lists were generated with the Mascot Distiller version 2.4.1 software (Matrix Science) from the LC-MS/MS raw data. Using the Mascot 2.4 search engine (Matrix Science), MS/MS spectra were searched against the homemade RNA-Seq library described above and a homemade list of contaminants frequently observed in proteomics analyses [77]. The evaluation of dehydrin expression dynamics was performed using SDS-PAGE and Western blot in several states of dehydration, such as moderate, D1 (50% RWC); severe, D2 (20% RWC); and fully dry, D3 (6% RWC). The evaluation of drought stress and sampling were performed according to Mladenov et al. (2015). The Western blots were performed in triplicates and analyzed using the Image Quant software (GE). The signal abundance of the immunodetected bands was normalized to the overall sum of all transferred proteins visualized by a reversible stain (MemCode, Pierce).

### 4.7. Statistics and Pathway Analysis

The multivariate data were summarized by principal component analysis (PCA) and hierarchical cluster analysis (HCA) in the MatLab (Mathworks, Natick, MA, USA) software according to standard algorithms. The input matrices for PCA contained the log2 transformed values of the normalized abundances of spectra derived from the Progenesis software. For HCA, the log2 transformed averaged fold-change values for each protein were used. Co-expression patterns between proteins and corresponding contigs were evaluated by the regression plot analysis of the log2 transformed averaged values of the fold changes of proteins and corresponding contigs. A functional enrichment analysis and visualization of the transcriptomics and proteomics data were performed with the ClueGO plugin [78] with Cytoscape software [79] using a Bonferroni-corrected *p*-value < 0.05 threshold. Subsequently, the Kyoto Encyclopedia of Genes and Genomes (KEGG) pathway files for selected processes were downloaded and combined in a network according to node interactions. Transcriptomics [14,19], proteomics, and metabolomics [11,19,20] data were mapped and visualized as the average log2 ratio for each treatment compared with the control using the Cytoscape software. A signal flow analysis in primary metabolism was calculated using the Pathway Signal Flow Calculator (PSFC) [80] plugin for Cytoscape. The network topology from the KEGG database containing sucrose accumulation; starch degradation and synthesis; carbon fixation; lipid, fatty acid, and branched amino acid degradation; glycolysis/gluconeogenesis; and the TCA cycle was condensed and edited according to the available omics data. Unidirectional edges were used according to pathway files. The flow propagation was visualized based on the signal passing from the network’s inputs to its outputs, from lower levels to higher levels, as described in the PSFC manual. The drought-induced changes in fatty acids, starch, branched-chain amino acids, and carbon fixation through RbcL were used as impute nodes, while sucrose accumulation was used as a sink node.

## 5. Conclusions

In the present study, we explored changes in the proteome of the resurrection plant *H. rhodopensis* upon desiccation using various extraction and enrichment protocols coupled with shotgun and gel-based approaches. We used TEM and previously published transcriptomics and metabolomics data to integrate these results with changes in subcellular organization and metabolism during the desiccation of this resurrection plant. These results may prompt future investigations regarding the accumulation of osmolytes, antioxidants, and/or protective proteins in chloroplasts, for example, the specific transport of dehydrins into organelles. Further analyses, including techniques such as CO_2_ labeling and tracing, are needed to elucidate how carbon flux in *H. rhodopensis* is regulated upon desiccation.

Two proteins related to cell wall plasticity—UDP-apiose/xylose synthase and pectin esterase—were found to accumulate in the dry state, thus contributing to the preservation of the cell walls. Our TEM observations suggested that the internal structure of some organelles was altered during desiccation. We found an increased phosphorylation of LHCP, which could be related with the observed thylakoid remodeling and disassembly of PSII supercomplexes. On the other hand, the overexpression and phosphorylation of OEE1 could be involved in stabilizing PSII altogether with the overexpression of ELIPs. We also observed the overexpression of a PSI reaction center protein subunit and the PetA protein, which facilitates cyclic electron flow around PSI. Several isoforms of PPO have been found to accumulate in chloroplasts and could be involved in the preservation of these organelles as well. Together with the constitutive and inducible expression of various LEA proteins, HSPs, and antioxidants, we outlined the potential role of the phosphorylation of dehydrins for transport in the chloroplasts of *H. rhodopensis* to protect this organelle.

Despite these protective mechanisms, we observed some damaged chloroplasts, which we associated with the accumulation of several transcripts related to autophagy and ER stress. The degradation and reutilization of damaged structures could supply carbon precursors for the accumulation of osmolytes, notably the de novo synthesis of sugars, to promote cell survival. We showed using signal flow analysis that the degradation of fatty acids and branched amino acids could provide carbons through succinate dehydrogenase and malate shunts in mitochondria. Nonetheless, cell death is avoided in *H. rhodopensis* during drought stress, as indicated by the decrease in metacaspase-4 and the full recovery of plants after rehydration.

## Figures and Tables

**Figure 1 ijms-23-08520-f001:**
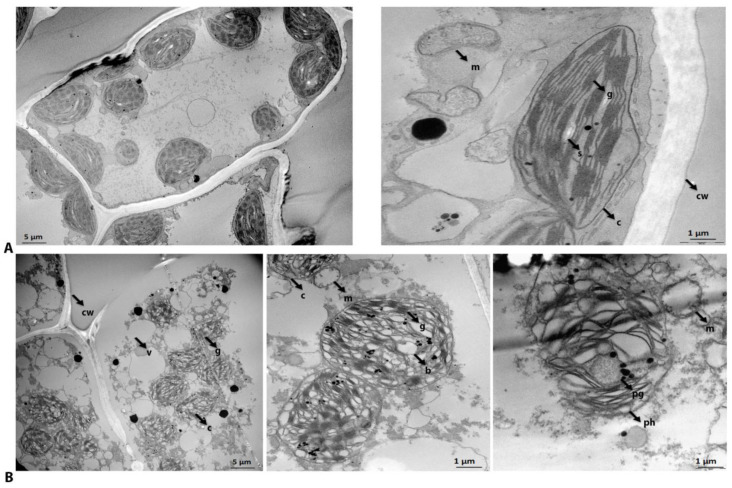
**Impacts of desiccation on the cellular ultrastructure of *H. rhodopensis* leaf cells**. Transmission electron microscopy leaf micrographs from fresh (**A**) and dry (**B**) plants. (**A**) Peripheral chloroplasts within the cell are shown in the left panel, and the internal organization of the cell can be observed in the right panel. c—chloroplast; g—granal thylakoid; s—starch granule; m—mitochondrion; cw—cell wall. (**B**) Chloroplast localization in the center of dry cells and fragmented vacuoles (v) is shown in the left panel. The internal chloroplast organization with granal thylakoids and bubble-like loculi (b), as well as mitochondria, can be observed in the middle panel, and the degradation of the chloroplast envelope in parallel to an accumulation of plastoglobules (pg) and features of phagocytosis (ph) in the chloroplast are shown in the right panel. Note the presence of a mitochondrion close to the chloroplast in the desiccated state.

**Figure 2 ijms-23-08520-f002:**
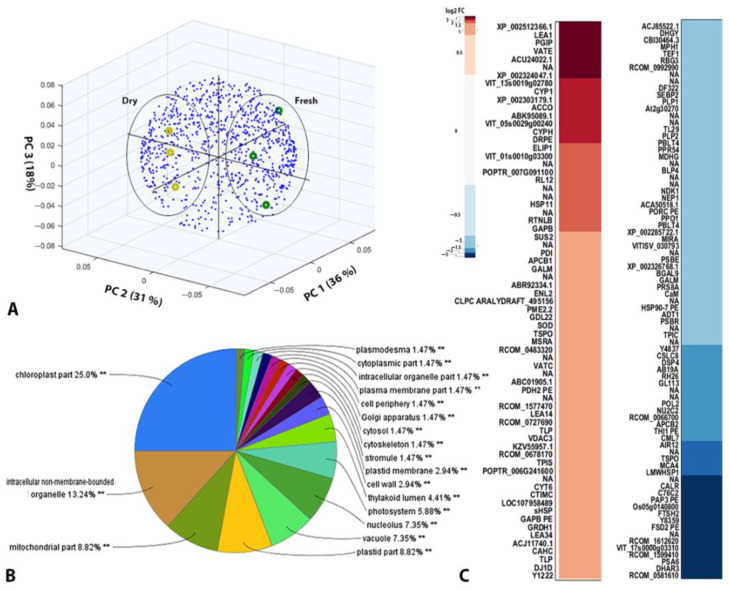
**Shotgun proteomics.** (**A**) Principal component analysis biplot showing the distribution of proteins (blue dots) from fresh and dry samples (colored circles) along with the first three principal components. (**B**) GO annotation of the subcellular localization of the identified proteins. Statistically significant enriched localization of identified proteins is given by asterisks (**C**) Heat map showing the mean log2 fold change (log2 FC) for the abundance of each identified protein in response to drought stress. Gene annotations are provided for significantly (|log2 FC| > 1) more (red) or less abundant (blue) proteins in dry plants.

**Figure 3 ijms-23-08520-f003:**
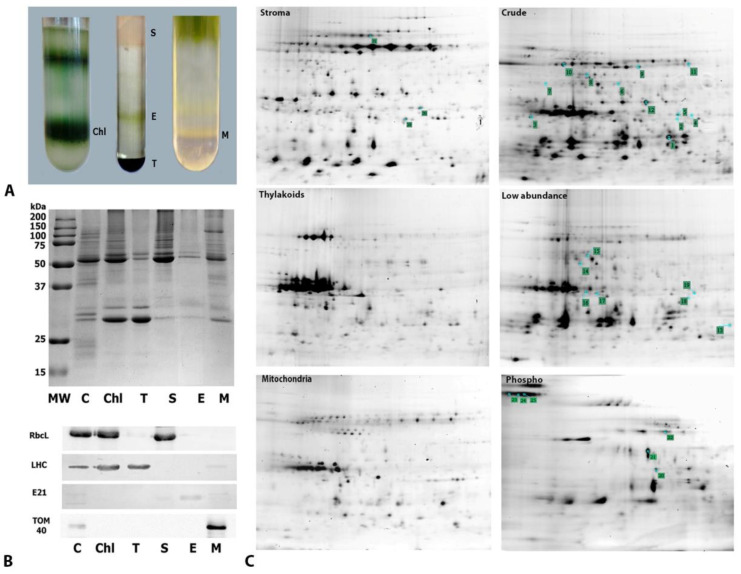
Extraction of intact organelles and their subcompartments from *H. rhodopensis* and gel-based proteomic analyses. (**A**) Purification of organelles and chloroplast subcompartments from *H. rhodopensis*. (Left) Percoll gradient allowing the separation of intact (bottom) and broken (top) chloroplasts (chl). (Middle) Percoll-purified chloroplasts fractionated into stromal (S), envelope (E), and thylakoid (T) subfractions using a sucrose gradient. (Right) Purification of the mitochondrial (M) fraction using a Percoll gradient. (**B**) SDS-PAGE analysis of organelle (upper panel) and suborganelle fractions and immunoblotting for the evaluation of cross-contamination (lower panel). MW—molecular weight marker; C—crude cell extract; Chl—chloroplasts; S—stroma; T—thylakoids; E,—envelopes; M—mitochondria. TOM40—translocator of the outer mitochondrial membrane 40; E21—chloroplast envelope protein 21; RbcL—RuBisCO large subunit; LHC—light-harvesting complex proteins. (**C**) Representative 2D-DIGE gels of the total protein extract (Total), low-abundance protein-enriched extract (Low abundance), phosphoprotein-enriched extract (Phospho), stromal fraction (Stroma), thylakoid fraction (Thylakoids), and mitochondria fraction (Mitochondria). Numbers represent the spots identified using matrix-assisted laser desorption/ionization time-of-flight/time-of-flight MS.

**Figure 4 ijms-23-08520-f004:**
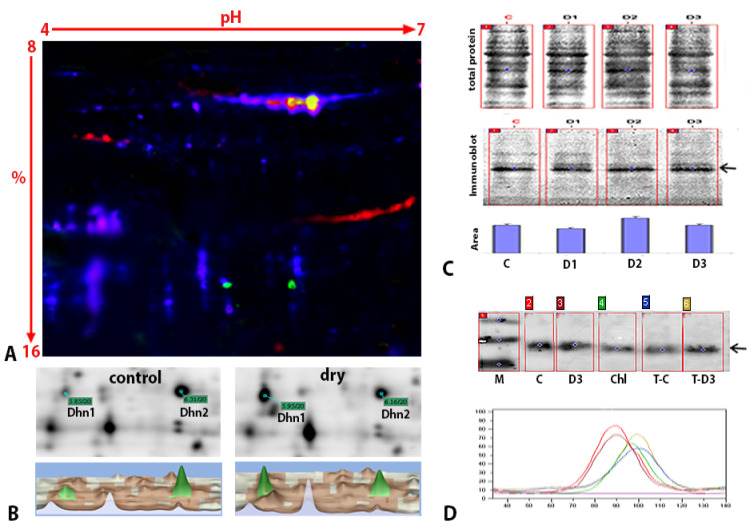
**Immunoblot detection and analysis of dehydrins in *H. rhodopensis*.** (**A**) Representative results from the 2D electrophoresis and immunoblotting of dehydrins. Proteins from a crude protein extract (mixture of internal standards for fresh and dry plants) were labeled with Cy5 (blue) and resolved using 2D electrophoresis with an isoelectric point (pI)/MW marker labeled with Cy2 (red). Subsequently, immunoblotting signals were detected with Cy3 (green). (**B**) Results of the 2D electrophoresis and quantitative analysis of proteins from fully hydrated and dry plants. (Upper panel) Representative gels. (Lower panel) 3D “landscape representation” of Dhn1 and Dhn2 from fully hydrated and dry plants with their corresponding relative abundances. (**C**) Immunoblot analysis of dehydrin expression at selected states of desiccation stress. C—watered plants; D1—moderate desiccation; D2—severe desiccation; D3—dry plants. (Upper panel) Resolved leaf protein extracts were stained with MemCode as a protein loading control. (Middle panel) Using anti-dehydrin serum, one 20 kDa protein band was detected (arrow). (Bottom panel) The mean area of each dehydrin band was normalized to the mean area of the total protein stain. Error bars were calculated from triplicate samples. (**D**) (Upper panel) Immunoblot detection of dehydrins in crude protein extracts and chloroplasts. C—watered plants; D3—dry plants; Chl—chloroplasts from watered plants; T-C—thylakoids from watered plants; T-D3—thylakoids from dry plants. (Lower panel) Colored numbers in each lane correspond to colors defining immunosignal areas.

**Figure 5 ijms-23-08520-f005:**
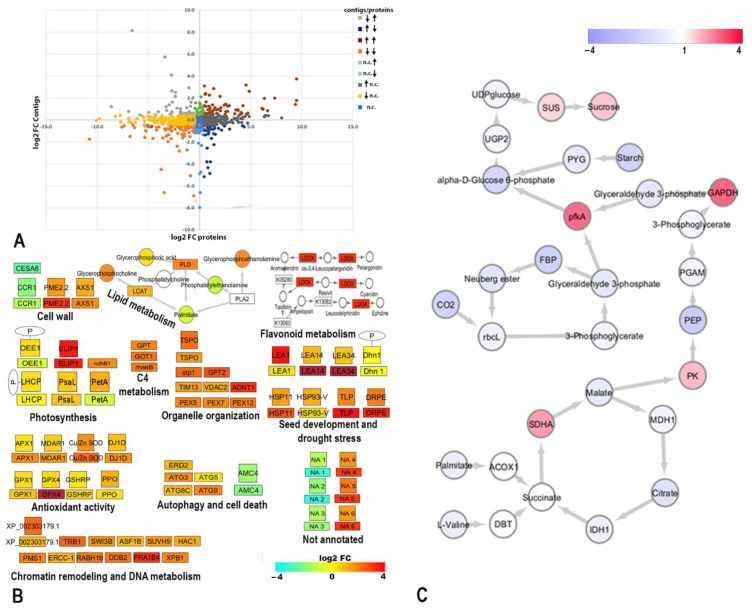
**Pathway analysis and co-expression changes in the proteome and transcriptome of *H. rhodopensis*.** (**A**) Scatter plot of averaged log2 transformed fold changes of abundances of RNA contigs (*x*-axis) and the corresponding proteins (*y*-axis). Colored bar represents significant expression changes above log2 of 0.5 for contigs and proteins (n.c.—no change; arrow up—upregulation; arrow down—downregulation). (**B**) RNA contigs and peptides annotated in the CLUE GO network were searched against the KEGG database and visualized together with available metabolomic data to clarify their involvement in cellular metabolism. Squares represent assigned proteins, rectangles represent transcripts, and circles represent metabolites. The colored bar represents changes in log_2_ FC between plants in the fresh and dry states. P—phosphorylation event. (**C**) Pathway signal flow analysis. The pathway was constructed by condensing and enriching a subset of reactions from several primary metabolism pathways of *A. thaliana* downloaded from KEGG. Unidirectional edges were used for reactions. Nodes are colored from blue to red, corresponding to low to high signals, respectively. The median was regarded as one because it represented the normal state.

**Table 1 ijms-23-08520-t001:** Expression, phosphorylation, and localization of identified proteins from different fractions. Each identified protein is annotated in the gels in Figure 2 with its corresponding spot number. The histograms for each spot display differences in spot volume between the fresh (left) and dry (right) samples (*n* = 3). Three-dimensional (3D) spot reports show the volume of each matched spot (green) in its corresponding organelle fraction.

Gel Annotation	Protein Score	Name	Gene	Crude	Phospho	Low Abundant	Stroma	Thylakoids	Mito
**2**	**177**	**Chlorophyll a-b binding protein 40**	**CAB40**	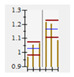	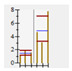	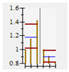	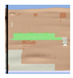	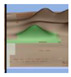	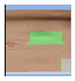
**3**	**122**	**Stem-specific protein**	**TSJT1**	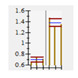	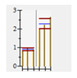	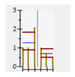	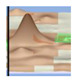	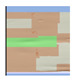	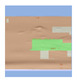
**11**	**179**	**ribulose bisphosphate carboxylase/oxygenase activase**	**RCA**	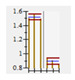	**N/D**	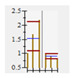	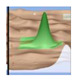	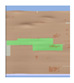	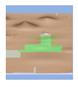
**18**	**222**	**Ribulose bisphosphate carboxylase large chain**	**RuBisCO**	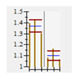	**N/D**	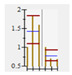	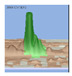	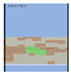	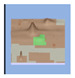
**12**	**95**	**Oxygen-evolving enhancer protein 1**	**OEE1**	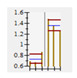	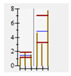	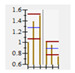	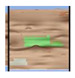	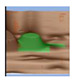	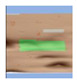
**13**	**136**	**Kunitz-type trypsin inhibitor B3**	**BPTI-3**	**N/D**	**N/D**	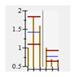	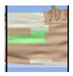	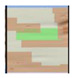	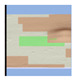
**14**	**240**	**quinone oxidoreductase-like protein**	**At1g23740**	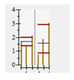	**N/D**	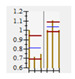	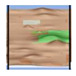	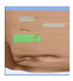	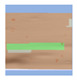
**15**	**177**	**phosphoglycerate kinase**	**PGKA**	**N/D**	**N/D**	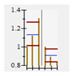	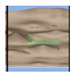	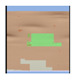	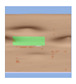
**16**	**150**	**proteasome subunit alpha type-2-A-like**	**L195_g004424**	**N/D**	**N/D**	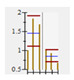	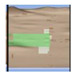	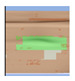	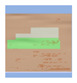
**17**	**331**	**N/A**	**N/A**	**N/D**	**N/D**	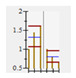	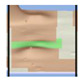	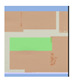	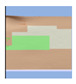
**20**	**131**	**dehydrin-like protein**	**Dh2**	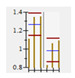	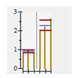	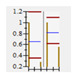	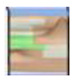	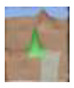	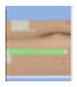
**21**	**219**	**hypothetical protein**	**N/A**	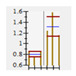	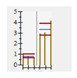	**N/D**	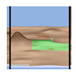	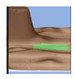	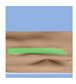
**22**	**86**	**hypothetical protein**	**N/A**	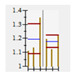	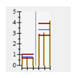	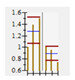	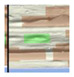	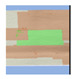	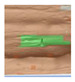
**23**	**86**	**hypothetical protein**	**N/A**	**N/D**	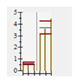	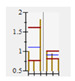	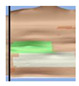	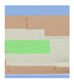	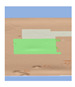
**24**	**273**	**hypothetical protein**	**N/A**	**N/D**	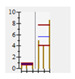	**N/D**	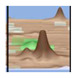	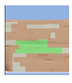	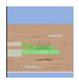
**26**	**211**	**polyphenol oxidase**	**PPO**	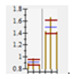	**N/D**	**N/D**	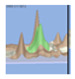	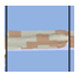	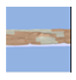

## Data Availability

The mass spectrometry proteomics data were deposited in the ProteomeXchange Consortium via the PRIDE partner repository with the dataset identifier PXD030496 and 10.6019/PXD030496.

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
