# Peer review of "Proteomics Evidence of a Systemic Response to Desiccation in the Resurrection Plant Haberlea rhodopensis"

_ijms, 2022, doi:10.3390/ijms23158520_

Round 1
Reviewer 1 Report
Resurrection plants are unique source of information regarding plant strategies for coping with harsh environments. In this respect, a comprehensive study combining two proteomic approaches (shotgun and gel-based which emphasize different subsets of proteins), subcellular fractionation, phosphorylation changes in specific proteins, and data interpretation in conjunction with TEM, transcriptomic and metabolomics data is of high value. Such a complex integrative study linking various omics data provides valuable new information. I strongly recommend acceptance of the ms for publication after minor revision.
Minor remarks:
Abstract
“to identify novel methods to improve resistance of crops to drought stress ” – not very clear, maybe - approaches instead of methods
“original large-scale proteomics” – proteomics is a quite well established technique, maybe - comprehensive, in-depth or systematic proteome analysis instead of original (with combination of various techniques)
Introduction – The first sentence is too general and could be omitted. A weak point here and in the discussion is not to refer to more recent studies on the ultrastructure, antioxidants and changes in the levels of various photosynthetic proteins in Haberlea upon desiccation detected by immunoblotting, which are in support of the authors` proteomic findings.
Results - It is not very clear for me how the use of 2D-DIGE-gel based approach could avoid difficulties related to the purification of intact chloroplasts from dried leaves. Gel based approach has it own advantages and it is very useful in detecting PTMs, isoforms of certain protein species, allows additional confirmation by immunoblotting, etc. But how It is related to purification of intact chloroplasts? Better justification is necessary. Maybe – the creation of a comprehensive 2DE proteome map of Haberlea using cell sub fractionation and enrichment in low abundance proteins.
Discussion – “The roles of HSPs, LEAs, and ELIPs in the drought stress response in resurrection plants have been comprehensively discussed elsewhere” – needs citation. Gechev et al. (2013) is actually ref [18].
MMs – “and spots that exhibited fold change of increasing or decreasing above 1.5 and p value <0.05 were targeted for picking and MS analysis. The spots with significant changes in the dry state and/or specific organelle localization were listed and targeted for excision by Ettan™ Spot Picker (GE Healthcare) “ – repetition in the two sentences, could be shortened
” For targeted analysis of dehydrins, …., were analysed.” – repetition, could be rephrased
References
27. – please provide journal, volume, pages (2013 Bulgarian J. of Agricultural Science 19(2):22-25)
72 – the same remark (Photobiochemistry and photobiophysics, 1986, 12 (3-4), 203-220)
Author Response
Thank you for your review. Please find our answers point by point:
- Abstract - “to identify novel methods to improve resistance of crops to drought stress ” – not very clear, maybe - approaches instead of methods
Answer: changed with approaches instead methods
- Abstract- “original large-scale proteomics” – proteomics is a quite well established technique, maybe - comprehensive, in-depth or systematic proteome analysis instead of original (with combination of various techniques)
Answer: changed with integrative instead original
- Introduction – The first sentence is too general and could be omitted. A weak point here and in the discussion is not to refer to more recent studies on the ultrastructure, antioxidants and changes in the levels of various photosynthetic proteins in Haberlea upon desiccation detected by immunoblotting, which are in support of the authors` proteomic findings.
Answer: First sentence is removed. Two new additional references related with cellular ultrastructure and antioxidants were added in Introduction and discussion: 18) Mihailova, G.; Vasileva, I.; Gigova, L.; Gesheva, E.; Simova-Stoilova, L.; Georgieva, K. Antioxidant Defense during Recovery of Resurrection Plant Haberlea Rhodopensis from Drought- and Freezing-Induced Desiccation. Plants (Basel) 2022, 11, 175, doi:10.3390/plants11020175.; 35) Georgieva, K.; Solti, Á.; Mészáros, I.; Keresztes, Á.; Sárvári, É. Light Sensitivity of Haberlea Rhodopensis Shade Adapted Phenotype under Drought Stress. Acta Physiol Plant 2017, 39, 164, doi:10.1007/s11738-017-2457-y.
- Results - It is not very clear for me how the use of 2D-DIGE-gel based approach could avoid difficulties related to the purification of intact chloroplasts from dried leaves. Gel based approach has it own advantages and it is very useful in detecting PTMs, isoforms of certain protein species, allows additional confirmation by immunoblotting, etc. But how It is related to purification of intact chloroplasts? Better justification is necessary. Maybe – the creation of a comprehensive 2DE proteome map of Haberlea using cell sub fractionation and enrichment in low abundance proteins.
Answer: 2D-DIGE-gel based approach is used also for organelle localization of differentially expressed spots by 2D DIGE software algorithm for intergel comparison using internal standard. Please find revised text in manuscript for better explanation.
- Discussion – “The roles of HSPs, LEAs, and ELIPs in the drought stress response in resurrection plants have been comprehensively discussed elsewhere” – needs citation. Gechev et al. (2013) is actually ref [18].
Answer: References are added.
- MMs – “and spots that exhibited fold change of increasing or decreasing above 1.5 and p value <0.05 were targeted for picking and MS analysis. The spots with significant changes in the dry state and/or specific organelle localization were listed and targeted for excision by Ettan™ Spot Picker (GE Healthcare) “ – repetition in the two sentences, could be shortened
Answer: Sentences are shortened
- ” For targeted analysis of dehydrins, …., were analysed.” – repetition, could be rephrased
Answer: repetition is rephrased
- References
27 – please provide journal, volume, pages (2013 Bulgarian J. of Agricultural Science 19(2):22-25)
72 – the same remark (Photobiochemistry and photobiophysics, 1986, 12 (3-4), 203-220)
Answer: references information is provided
Reviewer 2 Report
Good afternoon! I believe that this manuscript is a complete work. it can be published without changes. I won't detain you long and wish you all the best.
Please find more detailed comments in the attachments.

Author Response
Thank you very much for your kind comments and recommendation.
Corrections according to the other reviewer are accomplished.
All the best